# Environmental Impacts of Photovoltaic Energy Storage in a Nearly Zero Energy Building Life Cycle

**DOI:** 10.3390/ma15207328

**Published:** 2022-10-20

**Authors:** Rozalia Vanova, Miroslav Nemec

**Affiliations:** 1Department of Wood Structures, Faculty of Wood Sciences and Technology, Technical University in Zvolen, T. G. Masaryka 24, 96001 Zvolen, Slovakia; 2Department of Physics, Electrical Engineering and Applied Mechanics, Technical University in Zvolen, T. G. Masaryka 24, 96001 Zvolen, Slovakia

**Keywords:** timber construction, nearly zero energy building, attributional LCA, photovoltaic energy

## Abstract

Climate change, the economic crisis and the current geopolitical situation are the biggest challenges of today. They participate to a fundamental extent in the creation of international policies. Renewable energy sources are thus gaining worldwide popularity. The paper deals with the assessment of the impact of four selected stages of the life cycle of a NZEB building on the environment in 13 impact categories. The analysis is performed in accordance with the LCA method using the attributional modeling approach. The results show the partial and total shift of impacts on the environment of photovoltaic energy storage in comparison with photovoltaic energy export across the building life cycle. Along the climate change impact reduction as a positive effect on the environment, a substantial impact increase is observed on the depletion of abiotic resources. Results also show the total environmental impact of the building life cycle, considering the use of stored energy in a lithium-based battery as being beneficial in most categories despite the relatively high impact increment in the stage of replacement.

## 1. Introduction

Environmental awareness is increasing worldwide; sources of pollution are constantly being identified, and solutions are being sought to reduce them. International conventions are held, and new agreements are created, especially in the area of reducing the impacts of climate change [1,2,3,4,5]. In this context, the EU has set itself the goal of reducing net carbon emissions by at least 55% by 2030 compared to 1990 and achieving carbon neutrality by 2050 [6].

The construction sector is one of the areas covered by these goals. The report of the United Nations Environment Program [7] states “buildings accounted for 36% of global energy demand and 37% of energy-related CO_2_ emissions in 2020” and concurrently mentions reductions in emissions and energy intensity related to buildings by 17.2% to 48 kg CO_2_.m^−2^ and by 5.7% to 606 MJ.m^−2^ compared to 2015, respectively. In addition, the number of countries with the introduction of energy regulation of buildings increased by 30.6%. The COVID-19 pandemic has temporarily reduced global emissions, but it appears that we are still lagging behind the decarbonization of buildings, given the Paris Agreement commitments [8].

One of the options to reduce the negative impact of buildings on the environment is to increase energy efficiency. Currently, passive buildings represent a standard in the field of construction providing a satisfactory indoor environment in terms of thermal comfort and indoor air quality at the lowest possible energy costs. Passive building technology is based on a ventilation system with forced heat recovery, airtightness, improved thermal insulation and reduction in thermal bridges [9,10]. According to decree no. 364/2012 [11], family houses belong to the highest class of energy efficiency label A0 include buildings with a primary energy index ≤ 54 kWh.m^−2^.year^−1^, also referred to as nearly zero energy buildings [12]. These can be achieved by introducing renewable energy sources (RES). A study by Acaroğlu and Güllü [13] claimed that RES reduce emissions that are causing global warming. A study by Deymi-Dashtebayaz et al. [14] proved that an appropriately designed technological solution applying renewable energy sources can cover up to 70% and 57.9% of the electricity and heat demands, respectively. Therefore, a lot of authors appealing innovations and investment in energy efficiency technology remain significant in economic policy promoting [15,16,17,18]. A comprehensive review by Muteri et al. [19] thoroughly described the results and modeling approaches of different photovoltaic panel studies, highlighting electricity consumption during manufacturing stage as the main part of the environmental burden.

Life cycle assessment (LCA) provides a systematic assessment of the environmental aspects of the investigated system throughout its life cycle. ISO 14040 and 14044 establish the key principles, framework and procedures of the LCA method [20,21]. Within the construction industry, LCA enables to monitor and optimize the negative effects of either the construction materials themselves, the building as a whole or specific construction processes on the environment using indicators defining impact categories [16,22,23]. Rønning and Brekke [24] considered the identification of hidden problems related to the environmental load the main advantage of LCA.

Several software have been developed to improve the efficiency and reproducibility of LCA. Currently, among the most popular ones are SimaPro and Gabi, integrating a vast number of general databases and environmental impact assessment methods [25]. Yang et al. [26] highlighted the use of software to simplify LCA modeling and environmental impact analysis, allowing LCA practitioners to focus on basic data research and increase work efficiency. One of the most spread databases in Europe is the ecoinvent database [27], providing users with two basic modeling approaches—attributional and consequential. The attributional approach informs about the share of the environmental burden associated with the product within its life cycle at a given time, dividing the environmental burden of the process among the life cycles that this process serves, while using average data. In contrast, consequential modeling provides information on the environmental impact as a result of a decision that is related to changes in product demand and uses the approach of expanding the system with additional processes to deal with multifunctional processes, thus avoiding allocation.

Current LCA research in the field of NZEB covers two areas—the impact of construction materials (the production stage) and the operational energy impact related to the increase in energy efficiency (the operational stage). Climate change is the most assessed category among the number of other impacts [28]. From the construction material perspective, several authors have claimed that masonry buildings have higher embodied CO_2_ emissions contrary to wood-based buildings [10,29,30]. Authors [29,31] also demonstrated that it can affect the impact ratio between life cycle stages. Maierhofer et al. [10] assigned 22% of CO_2_ emissions to the building production stage and 67% of these emissions as being caused by operational energy requirements. Based on the study of Zhao et al. [32], photovoltaic energy can decrease impact on climate change up to five times compared to the electricity grid, highlighting the benefits of battery energy storage from the CO_2_ emissions point of view. On the other hand, they found mineral depletion to be alarmingly high.

NZEB are generally regarded as environmentally friendly buildings in the context of reducing greenhouse gas (GHG) emissions during the use stage of a building. The purpose of this study is to highlight impact categories that significantly affect the total impact of a selected NZEB within certain life cycle stages through the comparison of photovoltaic energy export and storage. The analysis is performed according to LCA principles using attributional modeling based on the cut-off system approach of the ecoinvent v3.8 database. The results may serve as a reference document for policymakers and stakeholders.

## 2. Materials and Methods

The system under study refers to a balloon-frame timber structure designed as a NZEB with a life span of 50 years (Figure 1). The two-story building has a gross floor area (GFA) of 210.0 m^2^ and gross internal area (GIA) of 159.6 m^2^ and is located in central Slovakia. The load-bearing structure consists of a finger-jointed solid timber insulated by rock wool. The foundations are designed with a concrete slab and foam glass filling. A flat roof structure is covered by external vegetation. Heating, ventilation and air-conditioning (HVAC) are performed via a heat-pump with controlled ventilation with recuperation, a wall-heating system and photovoltaic panels.

Table 1 and Table 2 show reference data used for the environmental assessment. The amounts of construction materials given in Table 1 originate from a bill of quantities of the considered construction. Values of travelled distances are based on a professional estimate and take into account the dimensions of the Slovak Republic as well as the approximate average distance of construction material transportation from the supplier to the construction site.

The B4 stage data comprise the necessary replacement of the HVAC worn parts based on a manufacturer’s recommendation. The data on operational energy are calculated according to the directive on the energy performance of buildings [12] and comply with the requirements on NZEB as the specific heat demand for heating of the construction equals 13.3 kWh.m^−2^.year^−1^. The estimated annual photovoltaic energy generation is calculated to 3956 kWh and exported to a grid. Both stages are modeled to reflect the listed energy and material consumption within 50 years of building usage.

Two life cycles are modeled, the one considering an export of the energy produced by a roof-placed photovoltaic system (Table 2) (the reference life cycle), and the other one for a storage of the energy in a lithium-based battery (the alternative life cycle).

As the photovoltaic energy is exported in the reference life cycle, the B6 stage is modeled through Equation (1):I_B6IC_ = I_EIC_ + I_HIC_ − I_PVIC_(1)
where I_B6IC_ is the impact of the B6 stage in selected impact category; I_EIC_ is the impact of electricity consumption in selected impact category; I_HIC_ is the impact of heat provided by the heat pump in selected impact category; and I_PVIC_ is the impact of photovoltaic energy in selected impact category.

The B6 stage in the alternative life cycle is assumed to utilize photovoltaic energy stored in a lithium-based battery (Equation (2)), and therefore, the amount of electricity needed from the grid is reduced.
I_B6IC_ = I_EIC_ + I_HIC_ + I_PVIC_(2)

In order to calculate input/output exchanges of the B6 life cycle stage related to the storage of the produced energy, the following battery properties are considered, namely, the energy from the photovoltaic system used in the building per year (E_U_), an efficiency factor (η) of 95%, a depth of discharge (h_v_) of 80% and losses (l) of 5% (Equations (3)–(5)):E_U_ = E_PV_ × η × h_v_ × (1 − l)(3)
E_U_ = 3956 kWh.y^−1^ × 0.95 × 0.8 × (1 − 0.05)(4)
E_U_ = 2856.2 kWh.year^−1^(5)
where E_PV_ is energy produced by the photovoltaic system per year.

Electricity losses (E_L_) (Equation (6)) thus equal to 1099.8 kWh.year^−1^.
E_L_ = E_PV_ − E_U_(6)

Subsequently, the need for electricity from the grid (E_G_) is 1285.6 kWh.year^−1^ (Equation (7)).
E_N_ = E_C_ − E_U_(7)

Exchanges of the lithium-based battery across the A1–A3, A4 and B4 stages are modeled using the following assumptions—battery weight of 50 kg and battery service life of 10 years.

The analysis is performed using SimaPro 9.3 (PRé Consultants, Netherlands) [33] software and the ecoinvent v3.8 database [27], using attributional system modeling based on the recycled content approach. Impact categories are chosen in accordance with EPD [34] and comprise four categories related to climate change, three categories of eutrophication, two categories of resource use, ozone depletion, acidification, photochemical ozone formation and water use. The impact results are compared between both variants of the life cycle, and categories reporting the biggest impact shifts are identified. The system boundary includes life cycle stages A1–A3, A4, B4 and B6 (Figure 2). Analyzed life cycle stages are selected in order to depict an impact change related to photovoltaic energy export and storage. End-of-life stages C1–C4 are omitted from the assessment, as these will be an object of thorough research.

## 3. Results

First, the environmental impact of the reference life cycle is set (Table 3). All the listed categories have a respective unit, whereas climate change categories have a common unit. The stage of transporting materials to the construction site (A4) has the lowest contribution of all the investigated stages. The B6 stage holds the highest environmental impact in almost all categories, except the photochemical ozone creation category, where the B6 stage impact (238.59 kg NMVOC eq) is slightly smaller than the product stage (A1–A3) impact (250 kg NMVOC eq). Moreover, the B6 stage is the only one utilizing minerals and metals resources (−0.41 kg Sb eq) in contrast to the stage of replacement (B4) constituting the highest load (1.98 kg Sb eq) of the given life cycle stages. The terrestrial eutrophication category indicates almost balanced impact across the A1–A3 and B6 stages (875.58 mol N eq and 879.52 mol N eq, respectively).

Second, the results of the alternative life cycle are compared with the reference life cycle. The following charts (Figure 3, Figure 4 and Figure 5) depict the change of a life cycle stage impact within impact categories (ΔI_SIC_) when energy storage is applied (I_SSIC_), compared to the impact of energy export (I_SEIC_) (Equation (8)).
ΔI_SIC_ = (I_SSIC_ − I_SEIC_)/I_SEIC_(8)

The biggest change in the impact of the product stage (A1–A3) is observed in minerals and metals use, accounting for 20.53% increment (Figure 3). The second highest increase in the impact of that life cycle stage occurs in the climate change impact category, and it does not exceed 5%. Next, freshwater eutrophication gains above 3% of the impact, followed by acidification transcending 2% line. The rest of the impact categories report even lower impact change.

The B4 stage shift of impacts within categories is much higher compared to the product stage (Figure 3). An increase of more than 35% of the impact is observed in the resource use, M&M and biogenic climate change categories, respectively. Two categories, namely, acidification and freshwater eutrophication, reach about 26% increment. Marine eutrophication, photochemical ozone creation and terrestrial eutrophication each gain above 15% of the impact. In five categories, the change of the impact ranges from 9% to more than 13% (water use; resource use, fossils; climate change, LULUC; climate change, fossil and climate change). Only the ozone depletion category is found to be minimal (1.80%).

The impact change caused by the transportation to the construction site (A4 stage) is negligible in nearly all categories (Figure 4). The highest increment is caused by LULUC emissions (0.02%). However, a reduction occurs in two categories, specifically in freshwater eutrophication and biogenic emissions of climate change of which the second mentioned reaches 0.09% cutdown of the impact and achieves the highest impact shift over all categories as well.

Notable impact changes occur in the B6 stage, of which the most remarkable is more than 663% decrease in minerals and metals use regarding the photovoltaic energy storage (Figure 5). In fact, it shows an increase of the resource use, considering that the B6 stage of the life cycle with the energy export is negative and equals −0.41 kg Sb eq (Table 3). The impact of water use and ozone depletion is higher by 35.61% and 24.36%, respectively. Other categories report an impact decline of which the highest shift ranges from about 42% to nearly 47% reduction in three categories, namely resource use, fossils; climate change, LULUC and freshwater eutrophication, respectively. Five categories reach lower impact of about 25% to nearly 32%, specifically marine eutrophication; acidification; climate change, fossil; climate change and climate change, biogenic, respectively. The lowest decreases of 15.26% and 7.81% of the energy storage impact are observed in terrestrial eutrophication and photochemical ozone creation.

Next, the impact within life cycle stages is summed and overall, the impact increment is set (Figure 6). The highest total impact change is observed in the minerals and metals resource use category (156.84%). Water consumption and ozone depletion increase by 18.43% and 8.47%, respectively. On the contrary, impact reduction occurs in most categories, especially freshwater eutrophication (−37.18%), fossils resource use (−30.30%), climate change LULUC (−27.06%) and climate change (−20.53%).

Subsequently, statistical analysis is performed, showing the mean, median, maximum and minimum change of the environmental impact of photovoltaic energy storage compared to the impact originating from the photovoltaic energy export (Table 4). The highest impact shifts come from the B6 stage (−663.73%), followed by the B4 (38.23%) and A1–A3 stages (20.53%), respectively. Stage A4 has a negligible impact at all. The highest impact shift in the A1–A3 life cycle stage is observed in the resource use of minerals and metals (20.53%), while the biogenic climate change emissions are nearly the same (0.00%). In the A4 stage, the biggest increase is noticed in the LULUC category (0.02%), whereas the main reduction comes from the biogenic carbon emissions (−0.09%). Minerals and the metals resource use category is mutually marked as the most affected for both B4 and B6 stages. The slightest impact change is observed in the ozone depletion category in the B4 stage (1.80%) and in the category of photochemical ozone formation in the B6 stage (−7.81%). To sum up, the use of the mineral and metal resources category is hit the most, resulting in more than a 150% increase in the impact demand. The least affected is the photochemical ozone formation category (−0.92%), and the highest impact reduction can be observed in the freshwater eutrophication category.

## 4. Discussion

The reference building life cycle deals with a wood-based construction in the standard of NZEB and reflects environmental impact related to photovoltaic energy export, which is compared to the impact of the alternative life regarding the storage of produced electricity in a lithium-based battery. Impact change in comparison with the reference wooden building occurs in the product stage, transport to the construction site, replacement and operational energy use. The building end-of-life stage is not taken into account.

### 4.1. Climate Change Impact Categories

The climate change category sums the emissions of three sources—biogenic, fossil and LULUC. The storage of energy increases CO_2_ emissions in the A1–A3, A4 and B4 stages and simultaneously decrease the impact of the B6 stage, resulting in overall impact reduction, except for biogenic carbon emissions. Due to the high quantity, fossil emissions mainly contribute to the reduction in the impact, despite the highest percentage impact decrease in LULUC emissions. Fossil emissions come from all fossil-burning processes over the course of the life cycle, with the energy sector being the major source [35,36].

Negative biogenic emissions reflect carbon capture, especially in the wood-based construction materials of the product stage [10,29,30]. Biogenic emissions in the B6 stage tend to lower as the share of electricity from the grid lessens.

### 4.2. Acidification, Eutrophication and Photochemical Ozone Formation Impact Categories

A negative impact change of emissions increase is detected in the A1–A3, A4 and B4 stages, followed by positive impact change, leading to emissions reduction in the B6 stage as well as the total emissions score. The lowest effect is connected with photochemical ozone formation. In contrast, percentage wise, the highest emission reduction occurs in freshwater eutrophication as the most positively affected category. Each category is connected with a supply of electricity from the grid [37], as it is supplemented by photovoltaic energy.

### 4.3. Ozone Depletion and Water Use

Emissions causing ozone depletion and use of water tend to rise across all of the studied life cycle stages resulting in the third and the second highest impact contribution in the overall score. The reason lies in the demanding photovoltaic panel production [38,39,40,41,42] intensified by the given modeling approach, as the impact of the reference life cycle in the B6 stage gives credit to the use of photovoltaic energy exported to the grid. Therefore, the impact of storage is burdened by the photovoltaic panel production impact [27].

### 4.4. Fossil, Minerals and Metals Resource Use Impact Categories

The impact of fossil resource use copies fossil climate change emissions within the life cycle. On the other hand, minerals and metals use is found to be the most negatively affected category in nearly all stages with a peak in total score of 156.84% increment. A change of nearly −664% in the B6 stage is derived from the actual negative impact of the reference life cycle (−0.41 kg Sb eq), thus resulting in an impact increase of 2.73 kg Sb eq to 2.32 kg Sb eq. Lithium together with cobalt, nickel and manganese, essential elements of battery systems, are regarded as critical raw materials [43] and play an important role in electronic devices manufacturing. An increased environmental impact related to the production of components of photovoltaic systems was proved by several authors [38,39,40,41,42].

### 4.5. General Discussion

NZEBs are designed as “green buildings” due to involving RES simultaneously with high energy efficiency [44,45]. The impact of construction materials supporting the energy efficiency, such as insulation or recuperation system, are part of the product stage. The operational energy use stage reflects the impact of direct energy requirements over the course of 50 years. Therefore, the outcomes of the study might be twofold as the effects of construction materials on the product stage and the B6 stage ratio, and the consequences of transition from energy export to photovoltaic energy storage on the environment.

Maierhofer et al. [10] studied the effect of masonry construction from the net-zero carbon targets perspective. According to the table (Table 5), construction materials influence the size of the product stage impact to a large extent, thereby also influencing the impact ratio between the A1–A3 and the B6 stages. From this point of view, wood-based buildings in the NZEB standard have a beneficial effect on the climate change impact regulation in the product stage. However, shifting the burden from the product stage to the operational energy use stage results in emphasizing the negative impacts of the B6 stage.

On the other hand, the ratio of 1:18 in the reference life cycle regarding photovoltaic energy export compared to the alternative life cycle regarding ratio of 2:23 declares energy storage reduces differences between the product and the operational energy use stages. Therefore, energy storage can be seen as one of the means of mitigating the impacts of climate change.

Energy storage reduces the amount of electricity needed from the grid, which in European conditions, comes mainly from fossil sources [36]. An impact reduction recorded in several categories—namely climate change, LULUC climate change, acidification, all three eutrophication categories, photochemical ozone creation and use of fossil resources—supports the findings of Gandiglio et al. [38] proving that RES reduce CO_2_ emissions compared to the fossil-based energy sources and at the same time escalate the water consumption. A mismatch is observed in the freshwater eutrophication category, where they found an increase in impact. This might be caused due to the different electricity grid mix [37].

High percentage increments sign a relatively low reference value as shown by Wu et al. [46]. Minerals and metals use is the major affected category in each analyzed life cycle stage, except the transportation stage in accordance with the study of Gandiglio et al. [38]. Climate change belongs to the most frequently analyzed impact categories, while abiotic resource depletion is one of the least investigated [47]. While multiple authors confirm difficulties with individual LCA study comparisons [19,23,47,48], more attention should be paid to create more studies concerning abiotic resource depletion. It is important to make this topic visible and to further analyze related aspects.

### 4.6. Limitations and Strengths

Results of the A1–A3 stage relate to a specific wood-based building structure. Other construction systems may have different impact results, as the wood components mainly influence carbon emissions [29,30,49,50] and thus influence the total score of a building’s life cycle, thereby distorting the benefits of energy storage. The B6 stage results relate to the impact of production, distribution and consumption of energies and refer to the national conditions of the Slovak Republic.

Databases used as a source data in the A1–A3 stage do not consider the transportation of the goods, while this feature is a part of the transportation stage (A4) covering average transportation distances in the Slovak Republic. Stages of replacement (B4) and operational energy use (B6) are bound to default distances given by the ecoinvent database [27]. Stages A1–A3 and B4 are applicable under the European production technology conditions.

For the application and comparison purposes of the study results, the same conditions must be considered, such as the type of used databases, the calculation method, the cut-off attributional system, and the geographical location.

## 5. Conclusions

The paper depicts the change in the impact of the building on the environment when storing photovoltaic energy in comparison with its export to the electricity grid in four stages of the building life cycle in the conditions of Slovakia. The reference building is designed according to the NZEB standard, using a balloon-frame structure and finger-jointed timber as the primary structural element.

Both positive and negative impact changes are observed between the individual life cycle stages of the building within the selected categories. To sum up, the main outcomes of the study are as follows:Impact of a vast majority of categories rises in the A1–A3, A4 and B4 stages when battery usage is applied;Positive effect of energy storage is observed in the B6 stage, where most of the categories reduce the impact;Adding up the impacts through the selected life cycle stages, the storage of photovoltaic energy in most categories reduces the overall impact on the environment, especially in the categories of climate change, fossil climate change, LULUC climate change, acidification, all three eutrophication categories, photochemical ozone creation and use of fossil resources.The most significant reduction in impact records freshwater eutrophication (−37.18%), fossils resource use (−30.30%), LULUC climate change (−27.06%) and climate change (−20.53%) categories, respectively.On the other hand, impact increase occurs in categories of biogenic climate change, ozone depletion, water use and use of minerals and metals. The last mentioned causes a significant burden of 156.84% impact increment, which can have a significant impact on the availability of these resources in the future and thus affect the economy. Therefore, it is important to take circular economy aspects into account, especially in connection with electrical devices usage.

The study showed that photovoltaic energy storage has several environmental benefits besides its climate change mitigation potential, underlining its justification in future applications. However, its massive mineral depletion potential pushes on the development of innovative circular economy strategies to avoid the eventual shortage of critical raw materials.

## Figures and Tables

**Figure 1 materials-15-07328-f001:**
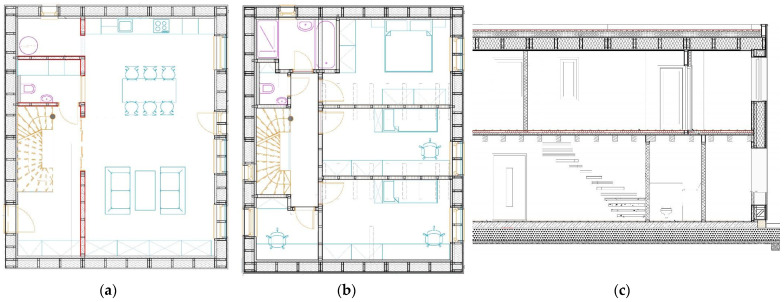
Scheme of the reference building: (**a**) ground floor plan; (**b**) first floor plan; (**c**) cross-section of the building.

**Figure 2 materials-15-07328-f002:**
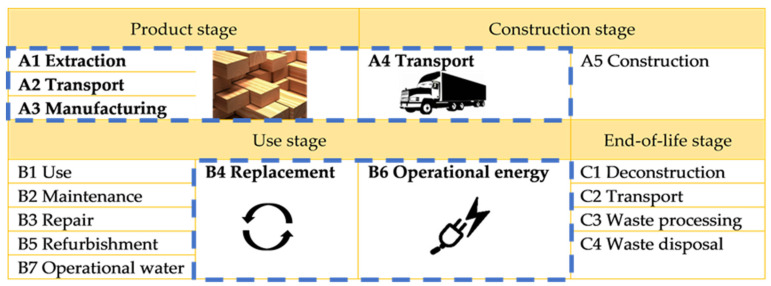
System boundaries.

**Figure 3 materials-15-07328-f003:**
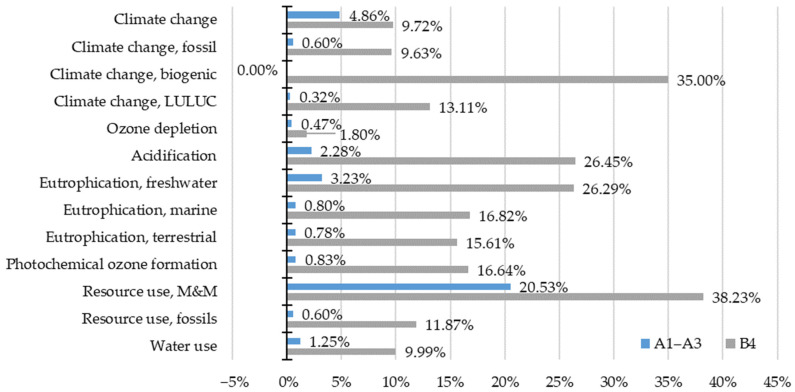
Environmental impact shift of A1–A3 and B4 stages considering photovoltaic energy storage.

**Figure 4 materials-15-07328-f004:**
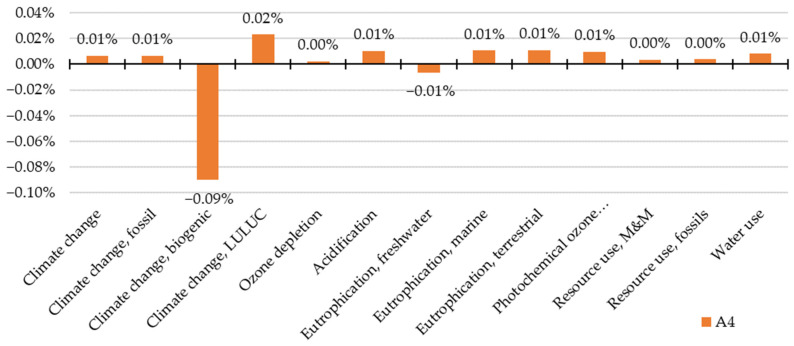
Environmental impact shift of the A4 stage considering photovoltaic energy storage.

**Figure 5 materials-15-07328-f005:**
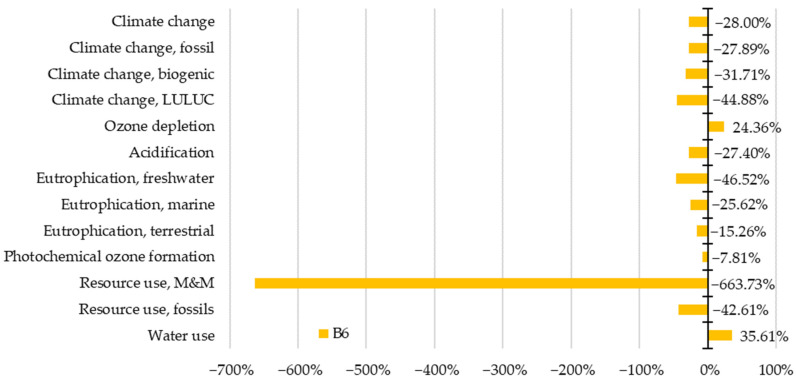
Environmental impact shift of the B6 stage considering photovoltaic energy storage.

**Figure 6 materials-15-07328-f006:**
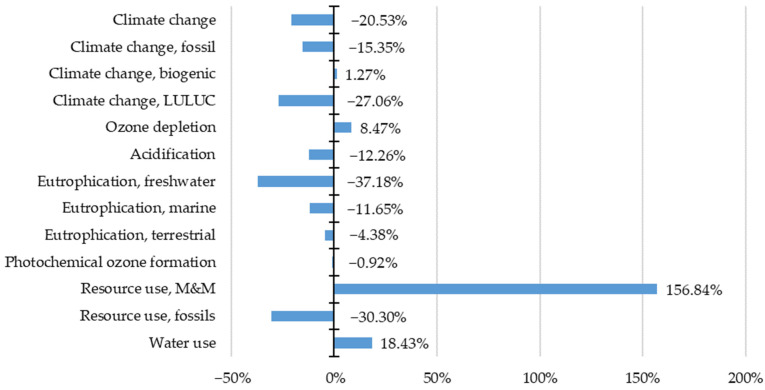
Total environmental impact shift considering photovoltaic energy storage.

**Table 1 materials-15-07328-t001:** List of materials included in the assessment. Columns Amount and Distance travelled (km) refer to stages A1–A3 and A4, respectively.

Material	Amount	Distance Travelled (km)
Timber	17,648.04 kg *	400
Oriented strand board	9881.80 kg *	100
High density fiberboard	3083.12 kg *	100
Gypsum plasterboard	8439.42 kg *	100
Rock wool	2245.89 kg *	100
Brick	6336 kg *	100
Reinforcing steel	1500 kg	200
Steel connectors	2331 kg	40
Gravel	23,720 kg	80
Concrete	45,210 kg	240
Foam glass	12,062.4 kg	200
Extensive vegetation	9074.8 kg	200
Triple-glazing wood-frame windows	4.42 m^2^ +18.84 m^2^	200 **
Inner doors	16.2 m^2^	200 **
Outer doors	2.88 m^2^	200 **
Mineral cover plaster	2934.5 kg *	40
Reccuperation system	200 kg	100
Refrigerants	57.4 kg	100
Scaffolding and accessories	162.7 + 40 kg	200 **
Packaging material	469.6 kg	-

* including 10% material surplus; ** considered import by one truck.

**Table 2 materials-15-07328-t002:** Data used to model B4 and B6 life cycle stages regarding photovoltaic energy export.

Stage	Input	Amount
B4	Filters in the recuperation system	100 pcs
	Heat exchanger	2.5 pcs
	Heat pump	2.5 pcs
	Photovoltaic panel	240 kg
	Refrigerant R134a	0.73 kg
	Propylene glycol	178.75 kg
B6	Electricity (Slovak electricity grid)	207.09 MWh
	Heat	290.33 MWh
	Photovoltaic electricity	197.8 MWh

**Table 3 materials-15-07328-t003:** Impact assessment of selected building life cycle stages regarding photovoltaic energy export.

Impact Category	Unit	Life Cycle Stage				Total
		A1–A3	A4	B4	B6	
Climate change	kg CO_2_ eq	6840.48	5099.21	20,264.71	119,336.79	151,541.2
Climate change, fossil	kg CO_2_ eq	55,559.55	5092.4	20,164.11	116,969.08	197,785.14
Climate change, biogenic	kg CO_2_ eq	−49,034.4	4.44	64.04	1955.42	−47,010.5
Climate change, LULUC *	kg CO_2_ eq	200.16	2.12	28	373.97	604.25
Ozone depletion	g CFC-11 eq	4.64	1.17	8.46	6.51	20.79
Acidification	mol H^+^ eq	338.1	20.57	154.08	735.8	1248.54
Eutrophication, freshwater	kg P eq	18.1	0.35	11.59	158.57	188.6
Eutrophication, marine	kg N eq	75.65	6.13	20.06	113.46	215.31
Eutrophication, terrestrial	mol N eq	875.58	66.98	238.03	879.52	2060.1
Photochemical ozone formation	kg NMVOC eq	250.89	20.55	67.78	238.59	577.81
Resource use, M&M	kg Sb eq	0.73	0.02	1.98	−0.41	2.33
Resource use, fossils	MJ	743,545.97	76,802.34	215,316.74	2,793,198.1	3,828,863.15
Water use	m^3^ depriv.	15,656.37	237.78	11,115.7	21,385.71	48,395.57

* abbreviation of land use and land use change.

**Table 4 materials-15-07328-t004:** Statistical analysis of the most affected impact categories.

Value	A1–A3	A4	B4	B6	Total
mean	2.81%	0.00%	17.78%	−69.34%	1.95%
max	20.53%	0.02%	38.23%	35.61%	156.84%
absolute max	20.53%	0.09%	38.23%	663.73%	156.84%
min	0.00%	−0.09%	1.80%	−663.73%	−37.18%
absolute min	0.00%	0.00%	1.80%	7.81%	0.92%
median	0.80%	0.01%	15.61%	−27.89%	−11.65%

**Table 5 materials-15-07328-t005:** Ratio comparison of climate change impact category based on m^2^ GFA.

Climate Change Impact Ratio of A1–A3 and B6	Share of A1–A3 Impact (%)	Source
1:18	5.55	Reference life cycle
2:23	8.70	Alternative life cycle
12:21	57.14	Maierhofer et al. [10]

## Data Availability

Not applicable.

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
