# Peer review of "Environmental Impacts of Photovoltaic Energy Storage in a Nearly Zero Energy Building Life Cycle"

_materials, 2022, doi:10.3390/ma15207328_

Round 1
Reviewer 1 Report
The authors investigated the impact of near zero energy consumption buildings with integrated photovoltaic modules on the environment in four stages of the life cycle through attributional modeling approach, and compared the impact of photovoltaic energy storage and photovoltaic energy export on different categories in the building, which is interesting. However, there are some questions that need to be addressed:
1. The introduction needs to be improved and innovations need to be strengthened to highlight the difference between the paper and previous work. The description of the energy background presentation ought to be shortened and more work related to the life cycle assessment of integrated photovoltaic buildings and ENZB should be supplied.
2. What are the data sources in Table 1 and Table 2? More details should be provided, such as the size of the building and the location of the building. (I believe photovoltaic power generation significantly affect by these conditions.) Please supplement the description and provide the structure diagram.
3. Assumptions made for the calculation model should be provided in the model part, not in the discussion part.
4. In the results and discussion part, the reasons for the differences between the alternative group and the reference group in various categories of indicators should be discussed, and the reasons for the impact of photovoltaic energy storage and export on indicators within the life cycle ought to be clarified.
5. In the conclusion part, the conclusions should be summarized by points to improve the logic
6. The reasons for ignoring the end-of-life stage of the building should be explained or provide references to support your approach
7. The full name of the abbreviation ought to be provided with the first appearance.
8. In line 33, the corresponding reference should be added for the data source.
9. In line 164, the subscript SEO does not appear above. Please check it.
10. In line 258, "The study is in compliance with...."This paper made a comparative study of the impact of photovoltaic energy storage and photovoltaic energy export. What is the impact of wooden based buildings base on the data in the investigation?
11. The formatting of the units should be corrected correctly, such as “Wh.m-2.year-1”, “MJ.m-2”
Author Response
Dear Reviewer,
thank you for your review, please find our responses in the attached file.
Best,
Ing. Rozália Vaňová, PhD.

Reviewer 2 Report
In my opinion, I find the study interesting and it only requires supplementing and clarifying a few issues.
1. What is the aim of the research?
2. Where do the data in the inventory table come from?
3. The values in the tables are not written correctly.
4. Charts should be editable.
5. Figure 4 value -663% requires a comment and explanation.
6. Summary is missing in the paper.
Author Response

(The authors gave the same response as above.)

Round 2
Reviewer 2 Report
Dear Authors,
The authors have significantly improved their typescript. I recommend accepting the typescript for publication.